# Three-Dimensional Organotypic Cultures Reshape the microRNAs Transcriptional Program in Breast Cancer Cells

**DOI:** 10.3390/cancers14102490

**Published:** 2022-05-19

**Authors:** Yarely M. Salinas-Vera, Jesús Valdés, Alfredo Hidalgo-Miranda, Mireya Cisneros-Villanueva, Laurence A. Marchat, Stephanie I. Nuñez-Olvera, Rosalio Ramos-Payán, Carlos Pérez-Plasencia, Lourdes A. Arriaga-Pizano, Jessica L. Prieto-Chávez, César López-Camarillo

**Affiliations:** 1Departamento de Bioquímica, CINVESTAV-IPN, Ciudad de México 07360, Mexico; yarely.salinas@cinvestav.mx (Y.M.S.-V.); jvaldes@cinvestav.mx (J.V.); 2Laboratorio de Genómica del Cáncer, Instituto Nacional de Medicina Genómica, Ciudad de México 14610, Mexico; ahidalgo@inmegen.gob.mx (A.H.-M.); mcisneros@inmegen.gob.mx (M.C.-V.); 3Programa en Biomedicina Molecular y Red de Biotecnología, Instituto Politécnico Nacional, Ciudad de México 07320, Mexico; marchat@gmail.com; 4Departamento de Biología Celular y Fisiología, Instituto de Investigaciones Biomédicas, Universidad Nacional Autónoma de México, Ciudad de México 04510, Mexico; blackvolkova@gmail.com; 5Facultad de Ciencias Químico Biológicas, Universidad Autónoma de Sinaloa, Culiacán 80030, Mexico; rosaliorp@uas.edu.mx; 6Laboratorio de Genómica, Instituto Nacional de Cancerología, Ciudad de México 14080, Mexico; carlospplas@gmail.com; 7Unidad de Investigación Médica en Inmunoquímica, Hospital de Especialidades del Centro Medico Siglo XXI, Instituto Mexicano del Seguro Social, Ciudad de México 06720, Mexico; landapi@hotmail.com; 8Laboratorio de Citometria de Flujo, Centro de Instrumentos, Coordinación de Investigación en Salud, Hospital de Especialidades del Centro Médico Siglo XXI, Instituto Mexicano del Seguro Social, Ciudad de México 06720, Mexico; lakshmi.litmus@hotmail.com; 9Posgrado en Ciencias Genómicas, Universidad Autónoma de la Ciudad de México, Ciudad de México 03100, Mexico

**Keywords:** breast cancer, 3D cultures, microRNAs, miRNA/mRNA coregulation networks

## Abstract

**Simple Summary:**

Three-dimensional (3D) cell cultures have several advantages over conventional monolayer two-dimensional (2D) cultures as they can better mimic tumor biology. This study delineates the changes in microRNAs (miRNAs) expression patterns of breast cancer cells cultured in 3D and 2D conditions. 3D organotypic cultures showed morphological changes such as cell–cell and cell–extracellular matrix interactions associated with a loss of polarity and reorganization on bulk structures in both basal Hs578T and luminal T47D breast cancer cells. Data indicate that downregulated miRNAs in Hs578T 3D cultures, relative to the 2D condition, contribute to a positive regulation of biological processes such as response to hypoxia and focal adhesion, whereas overexpressed miRNAs were related to negative regulation of the cell cycle. Remarkably, the reprogramming of miRNAs transcriptional profile was accompanied by changes in the expression of key miRNA/mRNA coregulation networks, such as miR-935/*HIF-1A*, which correlated with the expression found in clinical breast tumors and predicted poor patient outcomes. These data have implications in our understanding of cancer biology and impact the miRNA/mRNA regulatory axes of cells grown in 3D cultures. Our data represent a guide for novel miRNAs candidates for functional analysis, including the response to therapy and biomarkers discovery in breast cancer.

**Abstract:**

The 3D organotypic cultures, which depend on the growth of cells over the extracellular matrix (ECM) used as a scaffold, can better mimic several characteristics of solid cancers that influence tumor biology and the response to drug therapies. Most of our current knowledge on cancer is derived from studies in 2D cultures, which lack the ECM-mediated microenvironment. Moreover, the role of miRNAs that is critical for fine-tuning of gene expression is poorly understood in 3D cultures. The aim of this study was to compare the miRNA expression profiles of breast cancer cells grown in 2D and 3D conditions. On an on-top 3D cell culture model using a basement membrane matrix enriched with laminin, collagen IV, entactin, and heparin-sulfate proteoglycans, the basal B (Hs578T) and luminal (T47D) breast cancer cells formed 3D spheroid-like stellate and rounded mass structures, respectively. Morphological changes in 3D cultures were observed as cell stretching, cell–cell, and cell–ECM interactions associated with a loss of polarity and reorganization on bulk structures. Interestingly, we found prolongations of the cytoplasmic membrane of Hs578T cells similar to tunneled nanotubes contacting between neighboring cells, suggesting the existence of cellular intercommunication processes and the possibility of fusion between spheroids. Expression profiling data revealed that 354 miRNAs were differentially expressed in 3D relative to 2D cultures in Hs578T cells. Downregulated miRNAs may contribute to a positive regulation of genes involved in hypoxia, catabolic processes, and focal adhesion, whereas overexpressed miRNAs modulate genes involved in negative regulation of the cell cycle. Target genes of the top ten modulated miRNAs were selected to construct miRNA/mRNA coregulation networks. Around 502 interactions were identified for downregulated miRNAs, including miR-935/*HIF1A* and miR-5189-3p/*AKT* that could contribute to cell migration and the response to hypoxia. Furthermore, the expression levels of miR-935 and its target *HIF1A* correlated with the expression found in clinical tumors and predicted poor outcomes. On the other hand, 416 interactions were identified for overexpressed miRNAs, including miR-6780b-5p/*ANKRD45* and miR-7641/*CDK4* that may result in cell proliferation inhibition and cell cycle arrest in quiescent layers of 3D cultures. In conclusion, 3D cultures could represent a suitable model that better resembles the miRNA transcriptional programs operating in tumors, with implications not only in the understanding of basic cancer biology in 3D microenvironments, but also in the identification of novel biomarkers of disease and potential targets for personalized therapies in cancer.

## 1. Introduction

Triple-negative breast cancer (TNBC) refers to a molecular subtype of breast cancer that lacks the expression of both hormone receptors, the estrogen receptor (ER) and the progesterone receptor (PR), in addition to lacking the human epidermal growth factor receptor 2 (HER2) [1]. TNBC represents approximately 15–20% of all breast cancers and is the most aggressive molecular subtype [2]. Due to the lack of these receptors, endocrine therapy or HER2-targeted therapy are ineffective for treating TNBC, so there is no alternative treatment for this breast cancer subtype [3]. One of the main reasons for therapy failure is that cancer biology and drug screening have been largely studied using conventional two-dimensional (2D) monolayer cell cultures in vitro that do not represent the spatial–temporal tissue physiology observed in human tumors in vivo. In contrast, 3D organotypic cultures use substrates containing extracellular matrix (ECM) proteins for growth and can therefore better mimic some features of tumor tissues. Therefore, 3D cultures represent an attractive alternative to study cancer biology because the geometric space and the ECM microenvironment where they grow result in heterogeneous exposure to oxygen and nutrients, as well as in other physical and chemical stresses, resembling those cellular characteristics occurring in solid tumors [4]. Remarkably, 3D organotypic cultures provide cell–cell and cell–ECM interactions which are not mimicked in 2D monolayer cultures, as well as other physical and chemical stresses, resembling those characteristics occurring in vivo [5]. Therefore, there is an urgent need to study the interactions between ECM components and tumor cells throughout 3D organotypic environments, not only to better understand the biology of TNBC, but also to contribute to the development of novel and effective therapies.

Non-coding RNAs (ncRNAs) have been shown to participate in the transcriptional and post-transcriptional regulation of gene expression and function as novel biomarkers and potential therapeutic targets. Among the most studied ncRNAs are the microRNAs (miRNAs), which are single-stranded tiny RNA molecules of 21–25 nucleotides in length that function mainly as negative regulators of gene expression in diverse human diseases, including cancer [6]. MiRNAs regulate gene expression at the post-transcriptional level by binding to complementary sequences in the 3′ untranslated region (3′ UTR) of hundreds of target messenger RNAs (mRNAs), resulting in mRNA degradation or translational repression into cytoplasmic P-bodies [7]. MiRNAs have been implicated in the initiation and progression of cancers and influence the response to anti-tumor therapy, leading to drug resistance [7,8,9,10]. Due to their high stability, miRNAs are also being tested as therapeutic agents for treatment in clinical trials in cancer patients [11].

Nevertheless, most of our knowledge about miRNAs’ functions is derived from studies performed in 2D cell cultures that lack the characteristics of the tumor microenvironment, such as the interactions with the ECM, and heterogeneous exposure to oxygen, metabolites, and nutrients observed in the different layers of 3D cultures. It has been shown that much of the genetic and transcriptional heterogeneity of breast cancer can be attributed to the composition of the tumor microenvironment, which its better mimicked in the 3D environments. Recent studies have analyzed changes in mRNA and protein expression that mediate cellular signaling in 3D cultures and reported a differential reprogramming of transcriptional profiles with implications in cancer therapy [5,12]. However, changes in the miRNAs’ repertoire that are critical for fine-tuning of gene expression remain poorly studied, mainly because of the lack of systematic studies encompassing the different subtypes of breast cancer cells grown in 3D models. Therefore, here, we set up 3D organotypic cultures to evaluate the morphologic and cellular features of the less-well-characterized triple-negative Hs578T and luminal T747D breast cancer cell lines. Additionally, we reported the miRNA expression profiles of Hs578T cells cultured on a 2D plastic substrate, as well as in a 3D organotypic model using geltrex as an ECM-enriched scaffold, and found hundreds of differentially expressed miRNAs in 3D cultures related to the invasive phenotype of the TNBC cells, which are very different to those previously reported in two different subtypes of breast cancer cells. Implications of the reprogramming of miRNAs’ transcriptional profiles and miRNA/mRNA coregulation networks in the context of cell biology and cancer hallmarks’ activation are discussed.

## 2. Materials and Methods

### 2.1. Cell Cultures

Triple-negative Hs578T (ATCC^®^HTB-126 ™) and luminal T47D (HTB-133 ™) human breast carcinoma cell lines were obtained from the American Type Culture Collection and maintained in Dulbecco’s modified Eagle’s minimal medium (DMEM) supplemented with 10% fetal bovine serum and penicillin-streptomycin (50 units/mL; Invitrogen, Waltham, MA, USA) at 37 °C in a humidified incubator with 5% CO_2_ atmosphere. For 2D cultures, cells were grown on conventional tissue culture plates (Corning, Corning, NY, USA).

### 2.2. Three-Dimensional Organotypic Cultures

For the establishment of 3D cultures, one day prior, the LDEV-free and growth factor-reduced geltrex containing a basement membrane matrix enriched with laminin, collagen IV, entactin, and heparin-sulfate proteoglycans (Geltrex; Thermo Fisher Scientific, Waltham, MA, USA) was unfrozen at 4 °C. Subsequently, the 24-well flat-bottom plates were coated with 120 μL of LDEV-free, growth factor-reduced geltrex and incubated at 37 °C for 30 min. Then, 3.2 × 10^4^ cells were resuspended in a 250 μL final volume and incubated at 37 °C for 30 min. Finally, 250 μL of 5% geltrex medium was added and 3D cell cultures were maintained for 6 days, with DMEM medium changes every 2 days.

### 2.3. Immunofluorescence Experiments in 3D Cultures

Briefly, 3D cell cultures were fixed in 4% formaldehyde in PBS 1X for 30 min at room temperature. Coverslips were incubated with 0.1% Triton X-100 for 3 min. Following washing with PBS 1X, cells were blocked for 40 min at room temperature with 0.2% BSA in PBS 1X, and incubated with Phalloidin 1X (Abcam, ab235138, Cambridge, UK) for 1 h at room temperature. Nuclei were counterstained with diaminophenylindole (DAPI; Sigma, St. Louis, MI, USA). Slides were mounted with Vectashield Hardset Mounting Medium (Vector Laboratories, Burlingame, CA, USA).

### 2.4. RNA Isolation

For 2D cultures, cells were plated in a 6-well tissue culture plate (Corning, 3516) supplemented with DMEM medium. RNA was collected 2 days after plating at 70% confluency using the RNeasy plus mini kit (Qiagen, Hilden, Germany) according to the manufacturer’s protocol. For 3D cultures, cells were plated in a 24-well tissue culture plate. As previously described, RNA was collected 6 days after plating using the RNeasy plus mini kit (Qiagen) with minor modifications.

### 2.5. microRNAs Microarray Analysis

Experiments of DNA microarrays were performed at the Instituto Nacional de Medicina Genomica, INMEGEN, Mexico. Briefly, the experimental steps were as follows: RNA was labeled with the flash tag biotin HSR kit (Genisphere, Hatfield, PA, USA) according to the manufacturer’s instructions for poly(A)-tailing. Labeled samples were then hybridized with Affymetrix GeneChip miRNA 4.0 Array (Affymetrix, Santa Clara, CA, USA). This microarray contains 30,424 total mature miRNA probe sets, including 2578 mature human miRNAs and miRNAs from 202 other organisms. Samples were washed and stained with the Affymetrix GeneChip2 hybridization wash and stain kit, and then scanned with the Affymetrix GeneChip Scanner 3000 7G to generate fluorescent images, as described in the manufacturer’s protocol. For miRNA array analysis, CEL-files of the raw data were produced with Affymetrix GeneChip command console software v4.0. Partek Genomics Suite software (St. Louis, MO, USA) was used for further analysis. The RVM *t*-test was used to filter the differentially expressed miRNAs because the test can effectively raise the degrees of freedom in cases of small samples [13]. *p*-values were adjusted for multiple testing using the Benjamini–Hochberg method with a false discovery rate (FDR) of 0.05. Fold changes were calculated by comparing gene expression levels between 3D and 2D culture samples, and then expressed as the ratio between the averages of normalized intensities of the two groups. miRNAs with a fold-change threshold > 2.0 were considered differentially expressed [14].

### 2.6. Target Gene Predictions

The target genes of the differentially expressed miRNAs were predicted using the algorithms as implemented in the miRTarBase portal (https://mirtarbase.cuhk.edu.cn/~miRTarBase/miRTarBase_2022/php/index.php, 26 November 2021) [15]. Experimentally validated (luciferase reporter gene assay, Western blot, and/or qRT-PCR) miRNA–target interactions were selected with the aim to obtain promising experimental evidence and to avoid overestimating the functions of deregulated miRNAs.

### 2.7. Enrichment Analysis

The DAVID database (http://david.abcc.ncifcrf.gov/, 9 February 2022) was used to perform gene ontology (GO) and Kyoto Encyclopedia of Genes and Genomes (KEGG) pathway analyses of differentially expressed miRNA target genes. The species was restricted to *Homo sapiens*, and the adjusted *p*-value (from the Benjamini–Hochberg method) of 0.05 was considered statistically significant [16,17]. GO terms included the following three criteria: molecular function (MF), cellular component (CC), and biological process (BP). Enriched GO terms were presented as enrichment scores. KEGG pathway analysis was performed to determine the involvement of target mRNAs in various signaling pathways. The R package clusterprofiler v3.8.1 was used for KEGG pathway annotation with the same criteria implemented in the bioconductor free online software (http://www.bioconductor.org/packages/release/bioc/html/clusterProfiler.htm, 11 January 2022) [18].

### 2.8. Analysis of the miRNA/mRNA Networks

Based on the 502 and 416 miRNA/mRNA interaction pairs predicted for down- and up-regulated miRNAs in 3D cultures, respectively, miRNA/mRNA pairs were filtered based on 10 signaling pathways (hypoxia, Wnt/β-catenin, cell cycle, apoptosis, TGFβ, JAK/STAT3, NOTCH, P53, PI3K/AKT/mTOR, and epithelial–mesenchymal transition) involved in cancer progression. Finally, the construction of the miRNA/mRNA coregulation networks was performed using Cytoscape v3.2.0 software (http://www.cytoscape.org, 25 April 2022) [19].

### 2.9. Statistical Analysis

Experiments were performed three times and results were represented as mean ± SD. One-way analysis of variance (ANOVA) followed by Tukey’s test were used to compare the differences between means. *p* < 0.05 was considered as statistically significant.

## 3. Results

### 3.1. Establishment of 3D Organotypic Cultures of Breast Cancer Cell Lines

To determine whether breast cancer cell lines cultured in the 3D microenvironment show changes in morphology with respect to 2D cultures, we comparatively analyzed cell lines corresponding to two breast cancer subtypes: (i) the poorly metastatic luminal T47D cell line, and (ii) the highly metastatic triple-negative basal B Hs578T cells. We established the so-called on-top 3D culture model described previously [20] with some modifications, as described in the Materials and Methods Section. Changes in the morphology of cells were examined by phase contrast microscopy (Figure 1). Results show that 2D and 3D cell cultures showed different morphologies. Hs578T cells grown in conventional 2D monolayers exhibited a mesenchymal, elongated, and adherent morphology on the plastic substrate, with little cell–cell contacts, characteristic of a highly metastatic cell line, whereas T47D cells formed a single layer on the rigid plastic substrate, spread horizontally, showing interactions with neighboring cells, as expected (Figure 1A).

In contrast, Hs578T cancer cells cultured in the 3D microenvironment formed spheroid-like stellate structures characteristic of the metastatic basal B subtype [12], so we observed characteristics of the epithelial–mesenchymal transition (EMT) process associated with invasion, whereas T47D cells also formed spheroid-like structures but the in form of masses with rounded colony outlines (Figure 1B). We then calculated the diameter of the 3D structures using the AMIDA program [21] and found that the diameter increased with respect to the incubation time. The average diameter of spheroids in T47D cells at day 6 was around 600 μm, whereas for Hs578T cells it was 500 μm (Figure 1C). In contrast, the number of 3D spheroid-like structures decreased over time (Figure 1D). In particular, the number of spheroids diminished dramatically in the Hs578T cell line at day 6. Interestingly, we also observed discrete prolongations of the cytoplasmic membrane of Hs578T cells contacting neighboring spheroid cells, such as the tunneled nanotubes previously reported in 3D-grown cancer cells [22], suggesting the presence of cellular intercommunication processes and the possibility of fusion between spheroids, which could explain, in part, the decrease in number and the increase in size of 3D structures over time reported here (Figure 1E,F).

### 3.2. The Morphology of 3D Organotypic Cultures of Hs578T Breast Cancer Cells

Subsequently, we analyzed the changes in morphology of Hs578T cells grown in 3D and 2D cultures using confocal microscopy. We focused on the alterations of filamentous actin (F-actin) because polymerization and depolymerization control the reorganization of the cytoskeleton, resulting in morphological changes. As shown in Figure 2 (upper panel), actin stress fibers of Hs578T cells cultured in 2D monolayers were observed as densely stained and extensive disorganized bundles, forming lamellipodia- and filopodia-like structures. Likewise, Hs578T cells cultured in 3D also displayed a similar morphology, with some differences to 2D cultures, where F-actin staining clearly showed the lack of cell–cell adhesion contacts whereas the counterstaining with DAPI showed the disorganized nuclei (Figure 2, middle and bottom panel).

### 3.3. Differential Expression Profiling of microRNAs in 3D Cultures Compared to 2D Cultures of Hs578T Triple-Negative Breast Cancer Cells

To understand the relevance of changes in post-transcriptional control of gene expression that are critical for fine-tuning signaling events, we performed an analysis of the global expression of miRNAs in 3D versus 2D cultures using DNA microarrays (Affymetrix Gene-Chip miRNA 4.0, *Homo sapiens*), as described in the Materials and Methods Section. A total of 354 differentially expressed miRNAs were detected in 3D organotypic cultures compared to 2D cultures of Hs578T cells. Of these, 205 miRNAs were significantly overexpressed (Appendix A), and 149 miRNAs were significantly downregulated (Appendix A) (log2 fold change ≥ 2.0, *p* ≤ 0.05). The signal intensity corresponding to the differential expression of miRNAs in the 2D and 3D groups is shown in the heat map in Figure 3A. We found at least seven clusters of differentially expressed miRNAs. Figure 3B shows the volcano plot for differentially expressed miRNAs between cultures in 3D compared to 2D. The horizontal axis represents the fold change, and the vertical axis represents −log10 (*p*-value). Data indicate that 354 miRNAs were differentially expressed in Hs578T cancer cells when grown in 3D conditions, evidencing the impact of cell–cell and cell–ECM interactions on gene expression associated with 3D cell cultures’ formation. The distribution of differentially expressed miRNA genes on chromosomes was analyzed through circus plots. Data showed that most overexpressed miRNAs are encoded in chromosomes 7, 11, 16, and 17 (Figure 3C,D). The top ten overexpressed and downregulated miRNAs, as well as their validated targets and functions, are listed in Table 1.

### 3.4. The Enrichment Analysis of Predicted Target Genes for miRNAs Deregulated in 3D Organotypic Cultures Using GO and KEGG Tools

To elucidate the roles of 3D-modulated miRNAs in the regulation of cancer hallmarks, the prediction of target mRNAs of the set of deregulated miRNAs was performed using miRTarBase, which is a database of experimentally validated miRNA/mRNA interactions. Subsequently, we used the target gene prediction data obtained using miRTarBase and performed gene ontology (GO) analysis and pathway enrichment analysis using KEGG to speculate on the possible functions of deregulated miRNAs in 3D cultures. GO analysis describes genes in terms of three functional groups: cellular component (CC), molecular function (MF), and biological process (BP). The most important GO processes ranked by their −log10 (*p*-value) enrichment score for each of the 205 overexpressed and 149 downregulated miRNAs in the 3D cultures are listed in Figure 4. For the repressed miRNAs, the most enriched GO terms in biological processes included genes involved in hypoxia response, catabolic processes, and cell cycle regulation. The most enriched molecular functions were ATP binding and chromatin binding, while the most enriched cellular components were focal adhesion and substrate cell-adherent junctions. These results suggest that downregulated miRNAs such as miR-5189-3p, miR-3160-1, miR-183-5p, miR-935, miR-615-3p, miR-20a-5p, miR-27a-3p, let-7a-5p, and miR-152-3p could exert positive regulation of genes associated with these processes, contributing to cancer development (Figure 4, left panel). However, the main biological processes predicted by the overexpressed miRNAs were cell cycle division and vesicle transport. The most enriched molecular functions were enzyme binding and chromatin binding. Finally, the most enriched cellular components were cytosol and Golgi apparatus membrane. These data suggested that overexpressed miRNAs, such as miR-6802-5p, miR-3175, miR-1247-3p, miR-6780b-5p, miR-7641, miR-6132, miR-1281, miR-4327, miR-4417, and miR-1290 could significantly contribute to a negative regulation of cell cycle and cell proliferation in 3D cultures (Figure 4, right panel). These data agree with previous reports, where it has been shown that in 3D cultures compared to 2D cultures, an oxygen gradient is generated, leading to a response to hypoxia, similar to what occurs in tumors.

Thereafter, a signaling pathways analysis was performed using the KEGG tool to identify pathways and molecular interactions related to the target genes of miRNAs. Twenty-five and twenty KEGG pathways were identified in the downregulated and overexpressed miRNAs, respectively. Our data showed that the downregulated miRNAs regulate genes such as: *hypoxia inducible factor 1 subunit alpha* (*HIF1A*), *protein tyrosine kinase 2* (*PTK2*), *mitogen-activated protein kinase 1* (*MAPK1*), *mitogen-activated protein kinase 8* (*MAPK8*), *KRAS proto-oncogene* (*KRAS*), *integrin subunit alpha 5* (*ITGA5*), *cell division cycle 6* (*CDC6*), *integrin subunit alpha 1* (*ITGA1*), *epidermal growth factor* (*EGF*), *phosphatidylinositol-4*,*5-bisphosphate 3-kinase catalytic subunit alpha* (*PI3K*), *matrix metallopeptidase 2* (*MMP2*), *signal transducer and activator of transcription 3* (*STAT3*), *signal transducer and activator of transcription 1* (*STAT1*), *transforming growth factor beta receptor 2* (*TGFBR2*), and *notch receptor 2* (*NOTCH2*), among others, involved in 25 signaling pathways, including MAPK kinase signaling pathways, focal adhesion, adherents’ junctions, actin cytoskeleton regulation, receptor–extracellular matrix interaction, and TGF-β (Figure 5, left panel). Moreover, overexpression of miRNAs significantly enriched genes such as: *cyclin-dependent kinase 6* (*CDK6*), *cyclin-dependent kinase 4* (*CDK4*), *N-acetyltransferase 1* (*NAT1*), *interferon regulatory factor 2* (*IRF2*), *SMAD family member 2* (*SMAD2*), and *tumor protein 53* (*TP53*), among others, involved in 20 signaling pathways, including pathways in cancer, p53 signaling pathways, cell cycle, ErbB, Notch, MAPK, and mTOR (Figure 5, right panel). These results suggest that 3D cell growth results in the modulation of the expression of a group of 354 miRNAs that have, as predicted targets, multiple genes involved in diverse signaling pathways crucial for cancer development and progression. Furthermore, KEGG analysis confirmed the results obtained by GO, suggesting that overexpressed miRNAs could promote the regulation of pathways involved in cell cycle arrest in viable quiescent layers of 3D cultures, while downregulated miRNAs could regulate hypoxia and focal adhesion pathways.

### 3.5. Networks of miRNA/mRNA Interactions in 3D Organotypic Cell Cultures

In order to evaluate the impact of changes in miRNA expression in the interactions with its target mRNAs, we constructed miRNA/mRNA coregulation networks from the 10 most downregulated miRNAs in 3D organotypic cultures. Data yielded the prediction of 502 miRNA/mRNA pairs with 10 nodes, represented by miR-5189-3p, miR-3160-1, miR-183-5p, miR-935, miR-181a-2-3p, miR-20a-5p, miR-25-5p, miR-615-3p, miR-152-3p, and miR-140-3p (Figure 6, top panel). For example, the network predicted that overexpression of *HIF1A* could result from the release of repression exerted by six downregulated miRNAs (miR-3160-1, miR-935, miR-5189-3p, miR-183-5p, miR-20a-5p, and miR-615-3p) in 3D cultures, thus promoting the response to hypoxia. Collectively, hypoxia results in upregulation of metastasis, resistance to chemotherapy, and a reduced survival rate [43]. These mechanisms lead to cancer progression. Moreover, the 10 miRNAs mostly overexpressed in 3D organotypic cultures resulted in the prediction of 416 miRNA/mRNA pairs with 10 nodes, represented by miR-4417, miR-1290, miR-7641, miR-4449, miR-1246, miR-6780b-5p, miR-3197, miR-3175, miR-1247-3p, and miR-6802-5p (Figure 6, bottom panel). For example, in this network, *CDK4/6* was predicted to be targeted by 3 miR-1247-3P, miR-3175, and miR-7641, and thus we could deduce that *CDK4* and *CDK6* expression might be downregulated in 3D cultures, promoting cell cycle arrest and maybe inhibiting cell proliferation in the quiescent inner layers of organotypic cultures. These data are in agreement with previous reports by Koedoot and co-workers, where they show that in 3D cultures of triple-negative breast cancer cells, there is an inhibition of cell cycle-related genes [5]. According to our network analysis, most of the targets connected with several miRNAs, suggesting that multiple miRNA/mRNA interactions have a combinatorial effect on gene regulation.

### 3.6. The Expression Levels of Several miRNAs Modulated in Hs578T 3D Organotypic Cultures Correlate with Expression Found in Tumors of Breast Cancer Patients

For an effective study of miRNA/mRNA coregulatory networks in breast cancer, it is important to select an optimal in vitro model in which the regulation of the pathways correlates as closely as possible to what occurs in primary tumors. Therefore, we first examined the expression of four miRNAs modulated in 3D cultures in different breast tumor subtypes using RNA sequencing data from breast cancer patients (Luminal = 370, HER2 = 25, TNBC = 81) and 76 adjacent normal tissue samples published in The Cancer Genome Atlas (TCGA) database. The data showed that there is a correlation of the expression patterns of the miR-935, miR-195, miR-140, and miR-25 modulated in 3D cultures compared to the expression patterns of breast cancer patients (Figure 7). Interestingly, we found a lower expression of the miR-935, miR-195, and miR-140 in 3D cultures compared to 2D cultures, and also in breast tumors subtypes compared to adjacent normal mammary tissues. These data suggested that expression of miRNAs modulated by the microenvironment reproduced in the 3D cultures resembles the expression levels found in tumors. However, the expression of some miRNAs downregulated in 3D cultures, such as miR-25, among others, did not match with breast cancer patients. This may be because 3D cultures have some limitations, for example, components of the tumor microenvironment such as fibroblasts and endothelial cells, among others, are missing. Subsequently, we analyzed the expression of some targets of the miR-935, miR-195, miR-140, and miR-25 in breast cancer tumors (Figure 8). Interestingly, *TGB1*, *AKT1*, *PTK2*, and *HIF1A* were overexpressed in patient biopsies compared to adjacent normal tissue, suggesting that miRNA/mRNA interactions may contribute to hypoxia response, induce invasion, cell migration, and resistance to therapy.

### 3.7. Low Expression of miR-935 and High Levels of Its Target mRNA HIF1A Correlate with Poor Patient Outcomes

To obtain clues about the potential clinical implications of miR-935/*HIF1A* axis dysregulation in triple-negative 3D cultures, we performed an overall survival analysis using Kaplan–Meier analysis in a cohort of breast cancer patients, as described previously [44]. Luminal, HER2+, and TNBC transcriptomic databases included overall survival clinical information from TCGA repositories. Transcriptomic data from luminal (*n* = 307), HER2+ (*n* = 105), and TNBC (*n* = 97) with a median follow-up of 33 months were used in this analysis. Results showed that low miR-935 levels (HR = 0.25, log-rank *p* = 0.02) and high *HIF1A* expression (HR = 1.53, log-rank *p* = 0.00076) were associated with poor overall survival of TNBC patients (Figure 9). In contrast, no significant association between miR-935 and *HIF1A* expression and overall survival in luminal and HER2+ breast tumor subtypes were found (Appendix A).

## 4. Discussion

The 3D organotypic cultures of cancer cells represent an attractive alternative to study tumor biology because the geometric space and the ECM where they grow result in heterogeneous exposure to oxygen and nutrients, as well as other physical and chemical stresses, resembling those cellular characteristics occurring in in vivo tumors. This diffusion-limited distribution of oxygen, nutrients, metabolites, and cell–ECM interactions is not mimicked in monolayer cultures [45]. It has been shown that much of the genetic and transcriptional heterogeneity of breast cancer can be attributed to the composition of the tumor microenvironment, mediated by cell–cell and cell–ECM interactions. Therefore, more and more efforts are being made for utilization of 3D tumor cell cultures in cancer biology research. Previous studies reported on the effects of 3D cultures in gene expression regulation in cancer cells [46,47,48,49]. In an early report, the alterations in cellular morphologies and mRNA expression patterns of diverse breast cancer cell lines grown in 3D conditions were published, which were associated with tumor cell invasiveness and metastases [12]. These findings indicated that the changes in gene and protein expression patterns were responsible for, at least in part, the distinct morphologies observed in the 3D microenvironment and that were highly specific for each subtype of breast cancer.

Here, reported data extended the initial findings on miRNAs’ expression in cancer cells grown in 3D cultures. Since the gene expression greatly depends on the breast cancer subtypes, the data presented here for a different and poorly characterized breast cancer cell line are novel and added an important piece in the puzzle of gene expression regulation governed by interactions of cancer cells with ECM-related proteins in a 3D microenvironment. Our data showed that miRNA profiles found in basal Hs578T cells were very different to those previously reported for luminal MCF-7 and triple-negative MDA-MB-231 cancer cells grown in 3D organotypic cultures [46]. In an outstanding report, Nguyen and coworkers showed that 49 and 28 miRNAs were modulated in 3D cultures of MCF-7 and MDA-MB-231 breast cancer cells, respectively [46]. Of the 149 miRNAs we found here downregulated in the Hs578T cell line grown in the 3D microenvironment, only 6 and 2 miRNAs were similar in MCF-7 and MDA-MB-231, respectively, whereas of the 204 miRNAs found overexpressed in Hs578T, only 3 miRNAs and 9 miRNAs were similar in MCF-7 and MDA-MB-231, respectively. Moreover, only 2 downregulated miRNAs and 3 overexpressed miRNAs in 3D cultures, respectively, were common in all 3 cell lines, indicating that the miRNA profiles are quite different and specific for each cell line. These data clearly exhibit the substantial differences in transcriptional and biological heterogeneity frequently observed in the three main subtypes of breast tumors. In other study, it was reported that 3D cell cultures stimulate the secretion of exosomes similar to the extracellular vesicles produced by in vivo tumors, highlighting the impact of 3D organotypic cultures, not only in gene expression, but also in cell–cell communication mechanisms [46].

Our data also revealed that the downregulated miRNAs significantly contributed to a positive regulation of biological processes, such as hypoxia response, catabolic processes, and focal adhesion. These results are in agreement with a previously published study, where it has been shown that a hypoxic environment is generated in the inner layers of 3D cultures [47]. Hypoxic conditions play a pivotal role in cancer progression through the regulation of cell migration, proliferation, and differentiation. Furthermore, hypoxia in the center of solid tumors drastically decreases the chemo-sensitivity of tumor cells, promoting resistance to therapy and leading to cancer progression [48]. Likewise, ECM components can interact with cell surface receptors, such as integrins and receptor tyrosine kinases. The crosstalk between integrins and growth factor receptors directly regulates downstream cell signaling and growth factor-induced biological processes, mainly proliferation, invasion, and resistance to therapy [49]. Subsequently, in miRNA–mRNA network construction, we observed that six downregulated miRNAs (miR-3160-1, miR-935, miR-5189-3p, miR-183-5p, miR-20a-5p, and miR-615-3p) presented *HIF1A* as a target, thus promoting the response to hypoxia. Collectively, hypoxia results in upregulation of metastasis, resistance to chemotherapy, and a reduced survival rate [47,48]. When we compared the expression of four miRNAs downregulated in 3D cultures with the expression in triple-negative breast cancer patients, we observed that the expression of miRNAs in 3D cultures presented a positive correlation with the expression in patients compared to the expression of miRNAs in 2D. These data suggest that miRNA expression profiles of 3D cultured cancer cells could match with tumor samples, corroborating previous reports that 3D cultured cells better mimic in vivo tissues compared to the 2D condition [46,47]. Finally, we focused on the clinical implications of miR-935, because previous studies have shown that miR-935 inhibits proliferation and invasion in glioma through the miR-935/*HIF1A* regulatory axis [27]. Likewise, inhibition of miR-935 has been shown to increase paclitaxel sensitivity through upregulation of SOX7 in non-small-cell lung cancer [50]. Interestingly, we found that miR-935 is downregulated in 3D organotypic cultures of the triple-negative breast cancer cell line Hs578T compared to 2D cultures and in triple-negative breast cancer patients compared to adjacent normal tissues. These findings suggest that miR-935 represents a potential target in triple-negative breast cancer progression.

## 5. Conclusions

In conclusion, we suggested that 3D organotypic cultures could represent a suitable model for studying gene expression regulation as they present some of the cellular and molecular characteristics of tumor tissues as opposed to traditional 2D cultures.

## Figures and Tables

**Figure 1 cancers-14-02490-f001:**
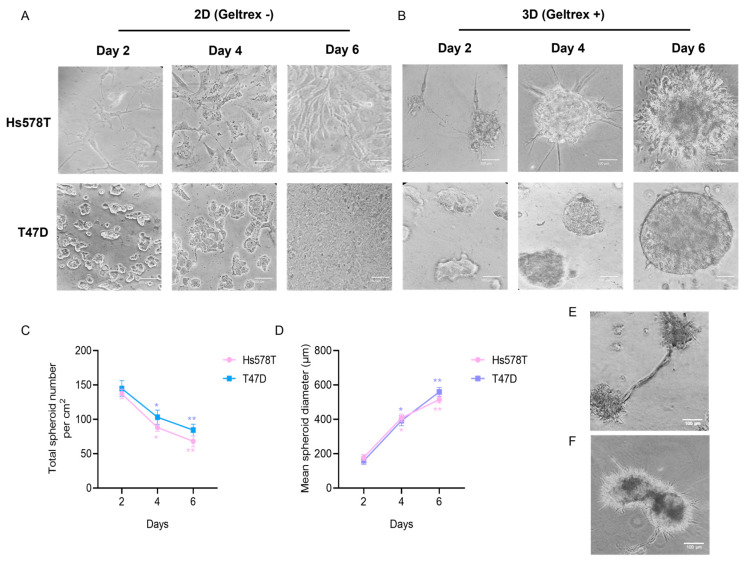
Establishment of 3D organotypic cultures of breast cancer cells. (**A**) Images taken by light microscopy (40×) corresponding to Hs578T and T47D cells cultured in 2D without geltrex for 2, 4, and 6 days. (**B**) Optical microscopy images (40×) of Hs578T and T47D cells cultured in the 3D microenvironment with geltrex reagent for 2, 4, and 6 days. (**C**) Quantification of the total number of spheroids formed in 3D cultures at 4 and 6 days in comparison to day 2, respectively. (**D**) Quantification of the average diameter of spheroids in 3D cultures at days 4 and 6 in comparison to day 2, respectively. (**E**,**F**) Light microscopy images (40×) showing nanotube-like structures (top panel) and probable fusion (bottom panel) between spheroids of Hs578T cells grown in 3D conditions. Scale bar = 100 µm. Data are representative of three independent experiments ± SD. * *p* < 0.05, ** *p* < 0.01.

**Figure 2 cancers-14-02490-f002:**
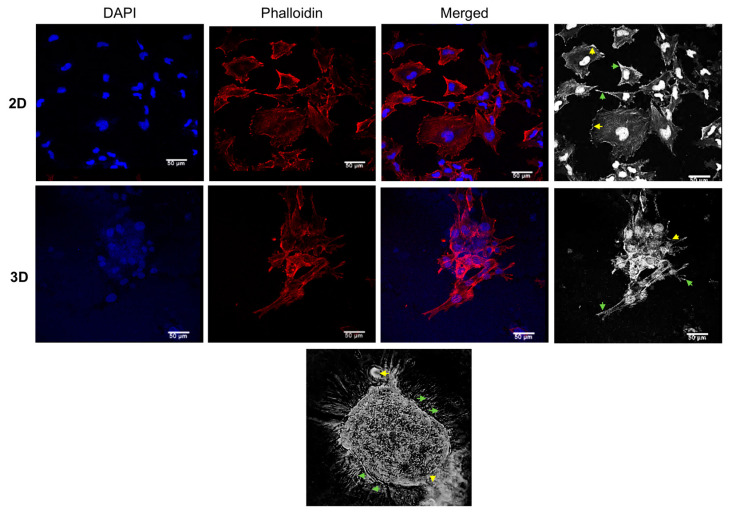
Morphology of Hs578T cells cultured in 2D and 3D cultures. Confocal microscopy images of 2D (**top panel**) and 3D (**middle** and **bottom panel**) cell cultures incubated with phalloidin-rhodamine for staining of F-actin stress fibers (red signal). Nuclei were counterstained with DAPI (blue signal). Lamellipodia-(yellow arrows) and filopodia-like (green arrows) structures. Scale bar = 50 μm.

**Figure 3 cancers-14-02490-f003:**
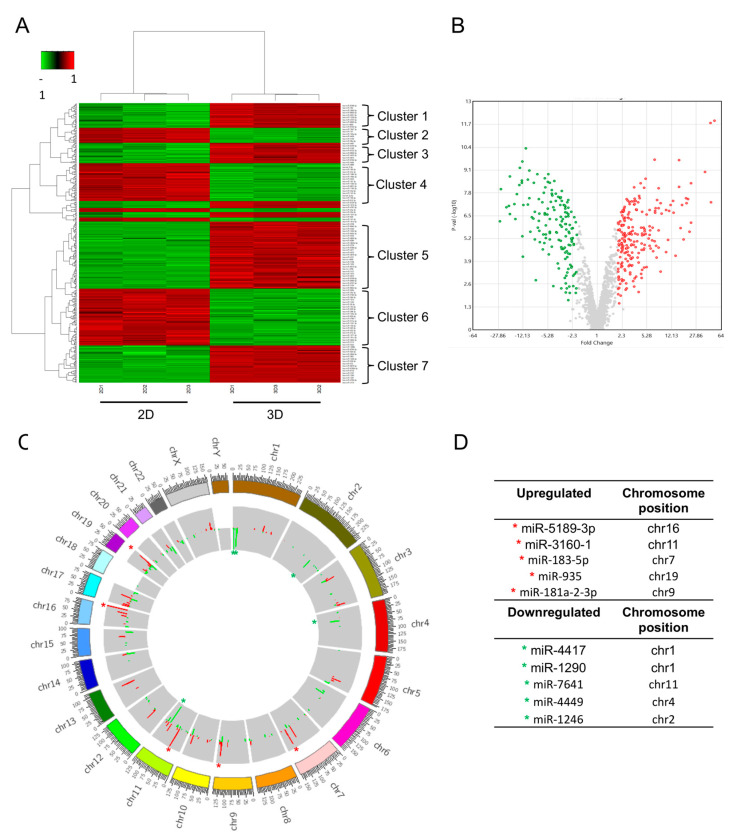
Microarray analysis of differentially expressed miRNAs in 2D versus 3D cultures of Hs578T breast cancer cells. (**A**) Heat map showing the signal intensity corresponding to differential expression of miRNAs in 2D and 3D cultures. Each column represents an individual sample, and each row represents a different miRNA. Unsupervised hierarchical clustering of differentially expressed miRNAs was performed using Pearson’s correlation coefficient. The expression levels of each miRNA in a single sample are represented according to the color scale. Red represents high expression levels, whereas green represents low expression levels. (**B**) Volcano plot graph represents the expression of miRNAs between 3D and 2D cultures. The *y*-axis represents the log2 ratio of the fold change and the *x*-axis represents −log10 (*p*-value). The red color indicates that the expression of miRNAs increased significantly more than 2-fold, whereas the green color indicates that the expression of miRNAs decreased significantly more than 2-fold in 3D organotypic cultures, compared to the 2D control group. (**C**) Circus plot. The circle graph represents the distribution of differentially expressed miRNAs in the chromosomes. The outer-layer circle denotes the chromosome map of the human genome. The inner layers represent the distribution of differentially expressed miRNAs on different chromosomes, respectively. The red and green colors represent overexpression and repression, respectively. (**D**) The five most overexpressed and downregulated miRNAs in 3D cultures, indicated in the circle graph with a red and green asterisk, respectively.

**Figure 4 cancers-14-02490-f004:**
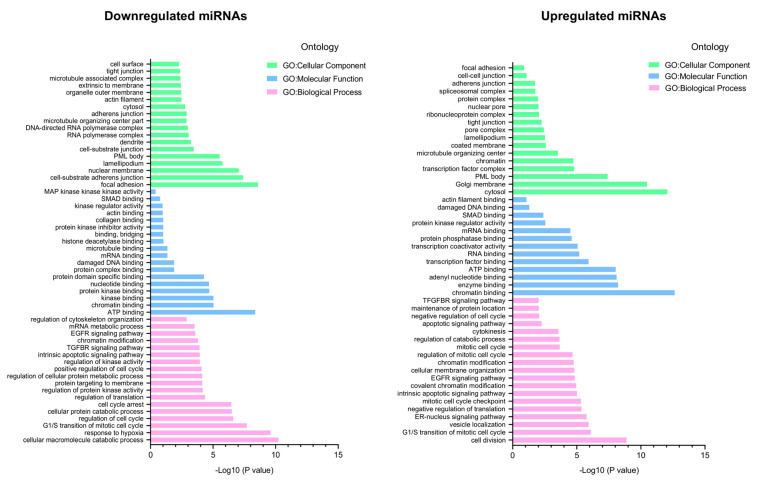
GO enrichment analysis of miRNA target genes deregulated in 3D organotypic cultures. GO annotation corresponding to downregulated (**left**) and overexpressed (**right**) miRNAs, including biological process (BP), cellular component (CC), and molecular function (MF).

**Figure 5 cancers-14-02490-f005:**
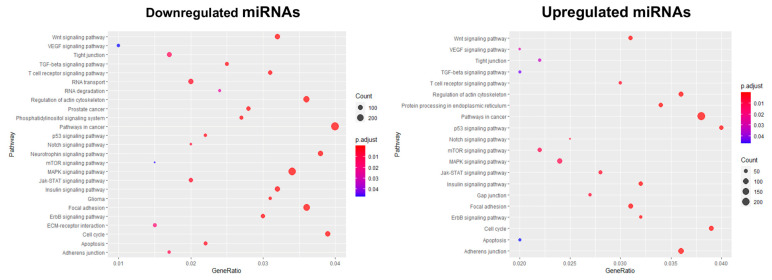
Signaling pathway analysis by KEGG for downregulated miRNA target genes in 3D organotypic cultures. Bubble plot of the KEGG enrichment analysis is shown, on the *y*-axis the enriched pathways of miRNAs downregulated in 3D organotypic cultures (**left panel**) and miRNAs overexpressed in 3D cultures (**right panel**) are shown. The size of the circle represents the number of genes involved in that pathway. The color represents the *p*-adjusted value.

**Figure 6 cancers-14-02490-f006:**
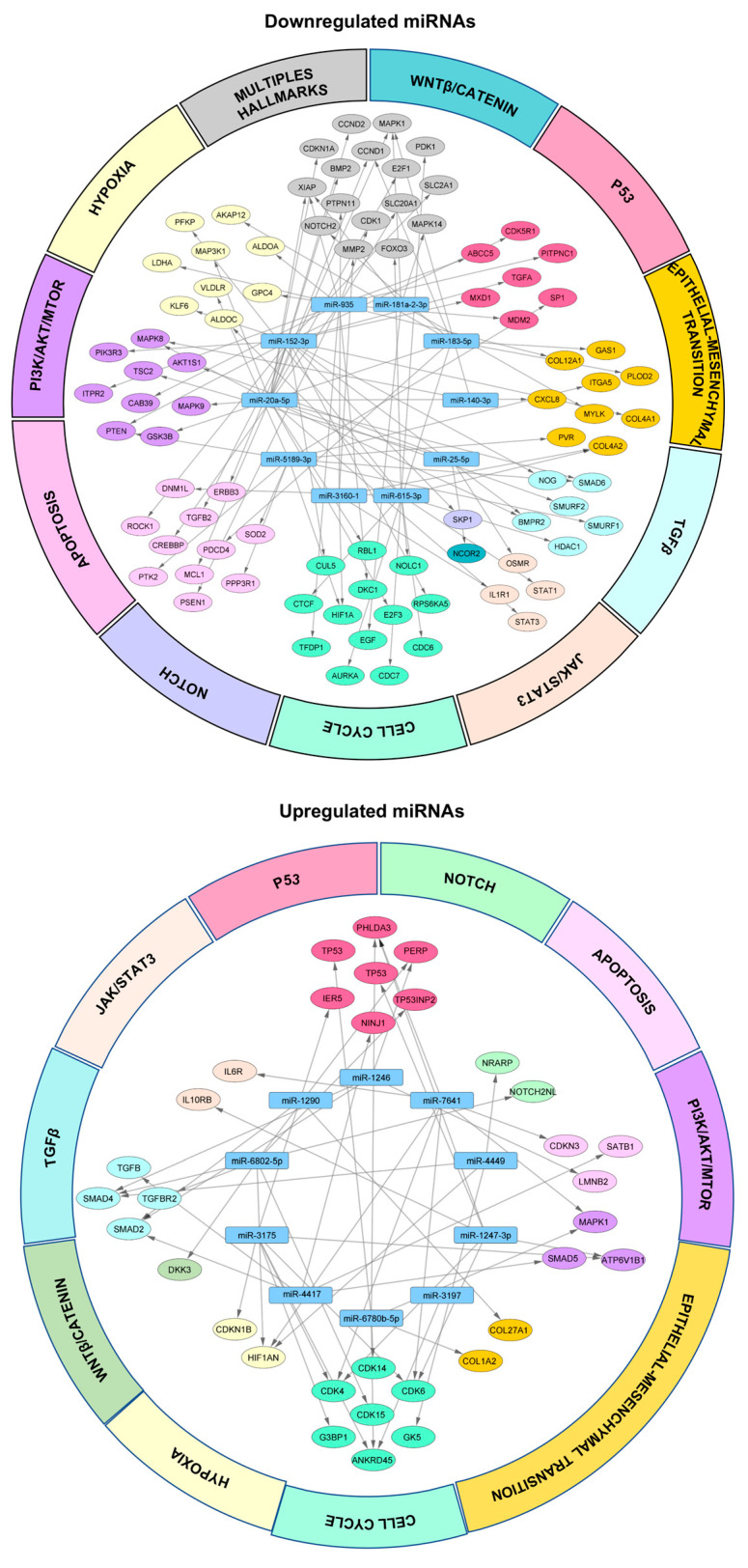
MiRNA/mRNA interaction networks of the 10 most deregulated miRNAs in 3D cell cultures. The graph shows the interaction network of the 10 most downregulated miRNAs in 3D organotypic cultures (**upper panel**) and the 10 most upregulated miRNAs (**bottom panel**) with their target mRNAs involved in hypoxia, Wnt/β-catenin, cell cycle, apoptosis, TGFβ, JAK/STAT3, NOTCH, P53, PI3K/AKT/mTOR, and epithelial–mesenchymal transition signaling pathways using Cytoscape software. The blue rectangle indicates the nodes represented by miRNAs, and the different colored ovals corresponding to each signaling pathway indicate the target mRNAs of the corresponding miRNAs.

**Figure 7 cancers-14-02490-f007:**
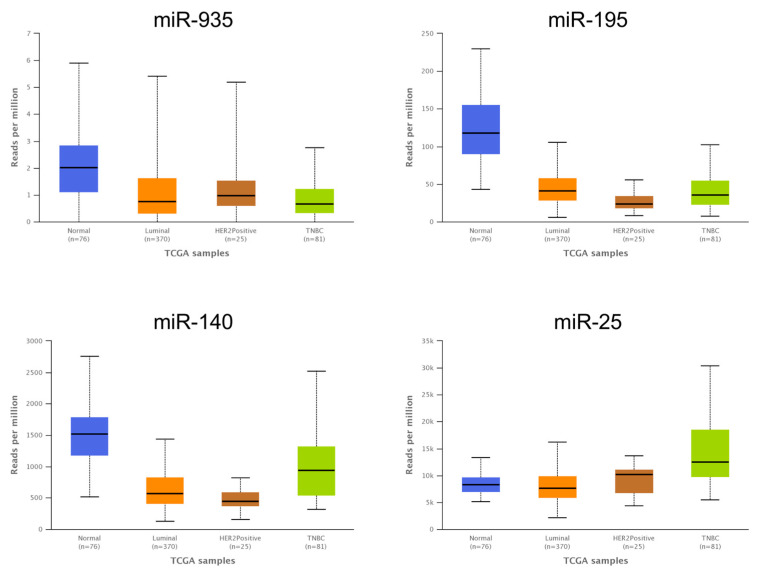
Comparison of the expression of four miRNAs downregulated in Hs578T 3D cultures against the expression reported in biopsies from breast cancer tumor subtypes according to TGCA databases. Bar graph showing expression levels of the four miR-935, miR-195, miR-140, and miR-25 in breast cancer subtypes compared to normal tissues (*n* = 76) from TCGA (RNAseq experiments). *Y*-axis shows reads per million according to RNAseq analysis. *X*-axis indicates the breast cancer subtypes (Luminal *n* = 370, HER2-positive *n* = 25, triple negative *n* = 81).

**Figure 8 cancers-14-02490-f008:**
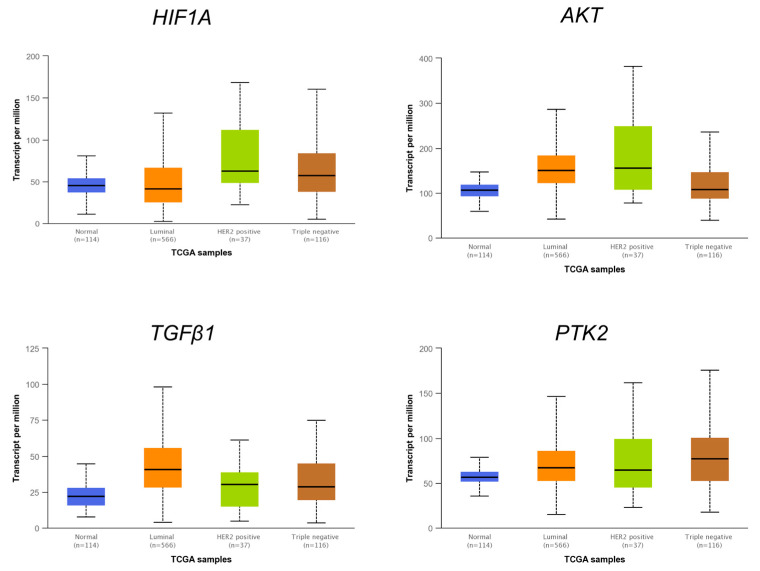
Expression of *HIF1A*, *AKT*, *TGFB1*, and *PTK2* mRNAs in biopsies from breast cancer tumor subtypes according to TGCA databases. Bar graph showing expression levels of the four mRNAs predicted as targets for the miRNAs downregulated in Hs578T 3D cultures. *Y*-axis shows transcripts per million according to RNAseq analysis deposited on TCGA. *X*-axis indicates the expression in breast cancer subtypes (Luminal *n* = 566, HER2-positive *n* = 37, triple negative *n* = 116) compared to normal (*n* = 114).

**Figure 9 cancers-14-02490-f009:**
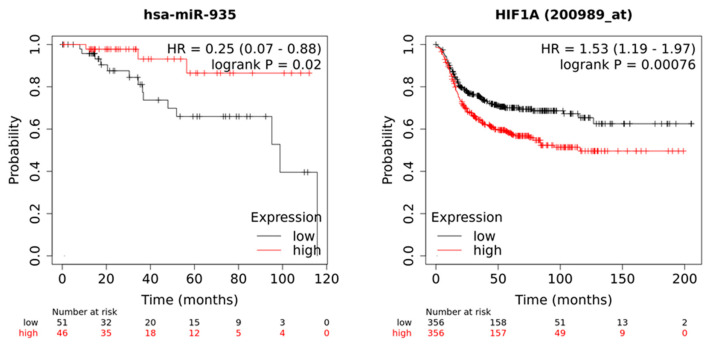
Kaplan–Meier curves for overall survival according to miR-935 and *HIF1A* expression. A cohort of triple-negative breast cancer patients with a median follow-up of 33 months was analyzed. Gene expression data and overall survival information were downloaded from TCGA. Patient prognostic values were obtained by dividing patient data into two groups according to various quantile expressions of miR-935 (miRpower tool 97 TNBC patients) and *HIF1A* gene (Start KM plotter, 712 TNBC patients). The two cohorts of patients were compared using a Kaplan–Meier survival plot, and the hazard ratio with 95% confidence intervals and log-rank *p*-value were calculated.

**Table 1 cancers-14-02490-t001:** Top 10 most downregulated and overexpressed miRNAs in 3D organotypic cultures compared to 2D cultures.

**Downregulated miRNA**	**Fold Change**	***p*-Value**	**Target**	**Function**	**Reference**
miR-5189-3p	−26.31	6.61 × 10^−7^	*JAG1*	Apoptosis	[23]
miR-3160-1	−25.52	9.72 × 10^−9^	unknown	unknown	N/A
miR-183-5p	−21	1.03 × 10^−7^	*FHL1*, *PDCD4*	Proliferation, metastasis, and angiogenesis	[24,25]
miR-935	−19.58	8.67 × 10^−8^	*SOX7*, *HIF1A*	Proliferation, invasiveness	[26,27]
miR-181a-2-3p	−18.72	2.62 × 10^−7^	*EGF*, *PI3K*, *SOX2*	Maintain cancer stem cell	[28]
miR-20a-5p	−18.66	1.80 × 10^−9^	*HMGA2*, *RUNX3*	Cell growth, cell mobility, and apoptosis	[29,30]
miR-25-5p	−16.26	6.75 × 10^−8^	*NEDD9*	Proliferation, invasion, and migration	[31]
miR-615-3p	−15.57	4.96 × 10^−7^	*PICK1/TGFBRI*	Promotes the EMT	[32]
miR-152-3p	−14.88	1.91 × 10^−8^	*EPAS1*, *PIK3CA*	Apoptosis, proliferation	[33,34]
miR-140-3p	−13.77	5.83 × 10^−9^	*TRIM28*	Cell growth, migration, and invasion	[35]
**Overexpressed miRNA**	**Fold Change**	***p*-value**	**Target**	**Function**	**Reference**
miR-4417	51.35	1.22 × 10^−12^	*TGF-β*; *SMAD2*	Migration and mammosphere formation	[36]
miR-1290	45.78	1.65 × 10^−12^	*NAT1*, *IRF2*	Proliferation and invasion	[37,38]
miR-7641	45.15	4.43 × 10^−9^	*ARID1A*	Proliferation	[39]
miR-4449	23.6	3.20 × 10^−8^	*SOCS3*	Proliferation	[40]
miR-1246	23.34	3.71 × 10^−8^	*CCNG2*	Migration and chemotherapy resistance	[41]
miR-6780b-5p	22.19	5.36 × 10^−8^	unknown	unknown	N/A
miR-3197	21.01	6.27 × 10^−8^	unknown	unknown	N/A
miR-3175	20.06	8.10 × 10^−7^	*PI3K*	Proliferation, invasion, and apoptosis	[42]
miR-1247-3p	19.83	2.64 × 10^−6^	unknown	unknown	N/A
miR-6802-5p	18.12	7.17 × 10^−6^	unknown	unknown	N/A

N/A, not applicable.

## Data Availability

Microarrays data is available at GEO (NCBI) accession number GSE203045.

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
