# Peer review of "Three-Dimensional Organotypic Cultures Reshape the microRNAs Transcriptional Program in Breast Cancer Cells"

_cancers, 2022, doi:10.3390/cancers14102490_

Round 1
Reviewer 1 Report
The manuscript describes a comparison of the culture of breast cancer cell lines in 2- and3-dimensional models. In particular, the authors investigate the expression of miRNAs and relate them to pathways involved in tumour progression. This is a generally well-written paper which adds to the literature supporting the use of 3-dimensional organotypic in vitro models as more representative of tumours.
Some grammatical errors in the text should be amended and minor queries are noted below:
In the title “Three dimensional 3D”, 3D is redundant
Abstract – “grown” should be “growth”
Abbreviations should be explained in text
Geltrex is not the same as Matrigel.
In Materials and Methods, can the authors explain why expression of morbidly obese and normal weight samples are mentioned?
In Figure 1 legend:
- p<0.05 is repeated
- are the statistics a comparison of the 2 cell lines? In 1D, there does not appear to be a difference between the cell lines at day 4.
- independent is one word.
In section 3.2 “(F-actin)” is repeated.
In Figure 3A, the authors mention 7 clusters of differentially expressed miRNAs. Could they describe on what basis they clustered?
Figure 3D is not referred to in text.
Text refers to Figure 6A and 6B, but the figure legend does not.
Figure 7 should be neater.
Author Response
We acknowledge to editors and reviewers 1 and 2 for the opportunity to revise the manuscript. Your critical suggestions that we have fully replied greatly increase the quality of our study. All amendments have been marked in yellow color in the revised text for your easy reference and reading. We have carefully reviewed the manuscript according the referee suggestions and provide a point-by-point response.
Reviewer 1
The manuscript describes a comparison of the culture of breast cancer cell lines in 2- and3-dimensional models. In particular, the authors investigate the expression of miRNAs and relate them to pathways involved in tumour progression. This is a generally well-written paper which adds to the literature supporting the use of 3-dimensional organotypic in vitro models as more representative of tumours.
Some grammatical errors in the text should be amended and minor queries are noted below:
In the title “Three dimensional 3D”, 3D is redundant
Reply: Thank you very much, for your very accurate comment. As reviewer suggested we have removed the repeated 3D word (Page 1, lane 2).
Abstract – “grown” should be “growth”
Reply: Thank you for your comment, we have made the correction (Page 1, lane 48).
Abbreviations should be explained in text
Reply: Thanks for the comments, we have added the meaning of the abbreviations throughout the manuscript.
Geltrex is not the same as Matrigel.
Reply. Thanks for the observation. As the reviewer suggested we have replaced matrigel to geltrex in the manuscript from. Although in the data sheets of both products, they have the same composition, we have selected the commercial name geltrex (Thermo Scientific) which is the specific product used here.
In Materials and Methods, can the authors explain why expression of morbidly obese and normal weight samples are mentioned?
Reply. Thank you very much for the observation. We have made the pertinent correction, and deleted the expression, since we are dealing with the expression of samples from 3D cultures and 2D cultures.
In Figure 1 legend:
- p<0.05 is repeated
- are the statistics a comparison of the 2 cell lines?
In 1D, there does not appear to be a difference between the cell lines at day 4.
Reply: Thank you to reviewer for pointed-out this important issue. We have deleted the repeated p value. About the statistics, we apologize for the confusion. The statistics showed in figure 1C and 1D correspond to the comparisons of number (fig. 1C) and diameter (fig. 1D) of spheroids for each independent cell line at days 4 and 6, in comparison to day 2 as control, respectively. We have corrected the figure 1 legend and added the missing P-values in the image for each cell line at different days in colors to avoid confusion.
We observed minimal, but significative (p<0.05), differences in the number (Fig. 1C) and diameter (Fig. 1D) of spheroids for Hs578t and T47D cells at day 4 relative to day 2. Also, we observed significative (p<0.01) differences for Hs578t and T47D at day 6 relative to day 2.
- independent is one word.
Reply: Thank you for your comments, corrections have been made.
In section 3.2 “(F-actin)” is repeated.
Reply: Thank you for your comments, corrections have been made (Page 7, lane 274).
In Figure 3A, the authors mention 7 clusters of differentially expressed miRNAs. Could they describe on what basis they clustered?
Reply: All miRNAs were clustered according to their similar expression levels defined using Euclidean distances.
Figure 3D is not referred to in text.
Reply: We appreciate the reviewer’s comments. As the reviewer suggested, the 3D figure has been mentioned in the text (Page 8, lane 307).
Text refers to Figure 6A and 6B, but the figure legend does not.
Reply: Thank you very much for the comment, we have redesigned the Figure 6, and replaced “figure 6A” and “figure 6B” legends by “upper panel” and “bottom panel”, respectively. Please note that this figure was changed as requested the reviewer 2 as the original figure seems to be complex and illegible for readers. We have filtered the data and showed the interactions miRNAs/mRNAs impacting the more important signaling pathways and cancer hallmarks. The new figure 6 is showed in the revised version of manuscript (Page 13).
Figure 7 should be neater.
Reply: Thank you very much for your wise comments. Now, figure 7 has been reorganized and spliced in two figures, 7 and 8, to make it more understandable. Also, we have rewritten the figure legends, simple as possible for readers.
Reviewer 2 Report
In this manuscript, Salinas-Vera et al. compared the miRNA expression profiles of breast cancer cell lines grown in 2D and 3D cultures. Specifically, the authors identified 354 miRNAs that were differentially expressed in 2D vs. 3D cultures of Hs578T BC line. Importantly, the authors observed that the upregulated and downregulated miRNAs were associated with negative regulation of cell cycle and promoting hypoxia/ focal adhesion responses respectively. Overall, the manuscript describes the importance of 3D cultures as a more representative breast cancer model that maintain miRNA transcriptomic programs compared to patient breast tumors. However, there are several points that remain unclear and needs to be clarified or addressed to strengthen the manuscript. These questions are detailed below.
- The use of organotypic culture in this manuscript is not appropriate as the cellular models were derived from relative homogenous cancer cell lines (T47D and Hs578T) with the lack of other cell types/ cancer cells of different transiting states. Moreover, the organotypic medium is not well described in the materials and methods. Did the authors use organoid medium as described in Sachs et al. 2018?
- The authors described the Hs578T cells grown in 3D presented an invasive phenotype based on F-actin and DAPI staining (Fig 2). This is an inconclusive statement to make without the demonstration of specific invasive markers that are commonly expressed by metastatic breast cancers e.g. vimentin, EMT markers etc. In addition, they should validate the retention of basal-like state (CK14) of Hs578T cells in 2D/3D.
- The methods of how the bioinformatic analyses were done and their associated statistical methods used were poorly described. The authors need to explain in detail on the analysis parameters. Sections 2.7 -2.9.
- Are there any statistical differences between the different breast cancer subtypes for the various miRNAs identified in the analysis? In addition, why the authors focus only on miR-935 for subsequent downstream analysis? This was not properly explain in the main text.
- Can the authors show the clinical implications of miR-935 in other breast cancer subtypes from TCGA cohort? Is the association of poorer survival by miR-935 specific only to TNBC patients?
Minor comments:
6. The abstract is too long. It is recommended that the authors to keep the abstract succinct and highlight only the important key messages to the readers.
7. Fig 6 is illegible, it is recommended to either reduce the number of hub genes by increasing the stringency of the analysis and to only show key interactions in the main figure.
8. Throughout the manuscript, the authors used repressed genes to represent downregulated/underexpressed genes. Repressed genes usually refer to switching off of genes. Please clarify and reword the main text if necessary.
9. All genes in main text/ figures should be in italics to differentiate them from proteins.
Author Response
We acknowledge to editors and reviewers 1 and 2 for the opportunity to revise the manuscript. Your critical suggestions that we have fully replied greatly increase the quality of our study. All amendments have been marked in yellow color in the revised text for your easy reference and reading. We have carefully reviewed the manuscript according the referee suggestions and provide a point-by-point response.
Reviewer 2
In this manuscript, Salinas-Vera et al. compared the miRNA expression profiles of breast cancer cell lines grown in 2D and 3D cultures. Specifically, the authors identified 354 miRNAs that were differentially expressed in 2D vs. 3D cultures of Hs578T BC line. Importantly, the authors observed that the upregulated and downregulated miRNAs were associated with negative regulation of cell cycle and promoting hypoxia/ focal adhesion responses respectively. Overall, the manuscript describes the importance of 3D cultures as a more representative breast cancer model that maintain miRNA transcriptomic programs compared to patient breast tumors. However, there are several points that remain unclear and needs to be clarified or addressed to strengthen the manuscript. These questions are detailed below.
- The use of organotypic culture in this manuscript is not appropriate as the cellular models were derived from relative homogenous cancer cell lines (T47D and Hs578T) with the lack of other cell types/ cancer cells of different transiting states. Moreover, the organotypic medium is not well described in the materials and methods. Did the authors use organoid medium as described in Sachs et al. 2018?
Reply: We appreciate your comments and apologize for the confusion. Our study was based on the methodology described by (Lee, G.Y. et al. Three-dimensional culture models of normal and malignant breast epithelial cells. Nat. Methods. 2007, 4, 359-365) which performs "on top" organotypic models by culturing cancer cell lines on top of a thin gel of solubilized extract derived from Engelbreth-Holm-Swarm mouse sarcoma cells (or matrigel) coated with diluted culture medium and 5% matrigel, as we described in Materials.
Regarding the organotypic cultures concept, we have adopted this concept and methods according to several studies performed by different authors which have described that organotypic models are in vitro systems constructed from commercial cancer cell lines and cultured on matrigel or a semisolid support, which simulates the extracellular matrix (Nguyen HT, et al 2012; Ferreira, L. P., et al 2021; Ren, G., et al 2021). Thus, here we, and others, have considered as suitable to adopt the same concept for breast cancer cells cultured in 3D conditions over matrigel.
Nguyen, et al (2012). The microRNA expression associated with morphogenesis of breast cancer cells in three-dimensional organotypic culture. Oncology reports, 28(1), 117–126. https://doi.org/10.3892/or.2012.1764
Ren, G., et al. (2021). Loss of Nitric Oxide Induces Fibrogenic Response in Organotypic 3D Co-Culture of Mammary Epithelia and Fibroblasts-An Indicator for Breast Carcinogenesis. Cancers, 13(11), 2815. https://doi.org/10.3390/cancers13112815
Ferreira, L. P., et al. (2021). Screening of dual chemo-photothermal cellular nanotherapies in organotypic breast cancer 3D spheroids. Journal of controlled release : official journal of the Controlled Release Society, 331, 85–102. https://doi.org/10.1016/j.jconrel.2020.12.054
Did the authors use organoid medium as described in Sachs et al. 2018?
The methodology described by Sachs et al. 2018 is based on obtaining organoids, from live tumor tissues of breast cancer patients and following by breast cancer cells isolation through a combination of mechanical disruption and enzymatic digestion. The disaggregated tumor cells are then seeded on extracellular matrix substitute basement membrane extract (or Matrigel) beads and covered with organoid culture medium. Organoids have been described as ex vivo systems, mainly patient-derived explants (PDE), are cultured on matrigel that simulates extracellular matrix as Sachs, and others, describe (Sachs et al. 2018, Xu H, et al 2018). Therefore, the main reason why we did not use the medium described by Sachs et al. 2018, is that they used for true breast cancer organoids, while in our work we established organotypic cultures from cell lines as described in literature.
Xu H, et al. Organoid technology and applications in cancer research. J Hematol Oncol. 2018;11(1):116. Published 2018 Sep 15. doi:10.1186/s13045-018-0662-9
- The authors described the Hs578T cells grown in 3D presented an invasive phenotype based on F-actin and DAPI staining (Fig 2). This is an inconclusive statement to make without the demonstration of specific invasive markers that are commonly expressed by metastatic breast cancers e.g. vimentin, EMT markers etc. In addition, they should validate the retention of basal-like state (CK14) of Hs578T cells in 2D/3D.
Reply: Thank you very much for the wise and critical comments. We agree with the reviewer about the lack of experimental data to support the conclusion about the existence an invasive phenotype in our model. Thus, we have changed the phrasing in the manuscript, in order to avoid overinterpretation of data. We have removed the statement about the “invasive phenotype”, and just describe the morphology of the cells observed in 2D and 3D cultures (Page 7, lane 286).
- The methods of how the bioinformatic analyses were done and their associated statistical methods used were poorly described. The authors need to explain in detail on the analysis parameters. Sections 2.7 -2.9.
Reply: Thank you very much for the invaluable observations. Now, the in-silico methodologies used here were better described and detailed (Page 5, sections 2.7-2.9), which improves the understanding of bioinformatic procedures.
- Are there any statistical differences between the different breast cancer subtypes for the various miRNAs identified in the analysis? In addition, why the authors focus only on miR-935 for subsequent downstream analysis? This was not properly explain in the main text.
Reply: Thank you for the comment, we have included a paragraph explaining why we focused on miR-935 in the discussion section (Page 16, lanes 561-569) as follows: “Finally, we focused on the clinical implications of miR-935 because previous studies have shown that miR-935 inhibits proliferation and invasion in glioma through the miR-935/HIF1A regulatory axis [50]. Likewise, inhibition of miR-935 has been shown to increase paclitaxel sensitivity through up-regulation of SOX7 in non-small cell lung cancer [51]. Interestingly, we found that miR-935 is down-regulated in 3D cultures of the triple-negative breast cancer cell line Hs578T compared to 2D cultures and triple-negative breast cancer patients compared to adjacent normal tissues. These findings suggest that miR-935 represents a potential target in triple-negative breast cancer progression”.
- Can the authors show the clinical implications of miR-935 in other breast cancer subtypes from TCGA cohort? Is the association of poorer survival by miR-935 specific only to TNBC patients?
Reply: Thank you very much for the comments, we agree with this interesting point. We performed again the Kaplan meier analysis including the 3 subtypes of breast cancer. We did not observe significant association between miR-935 an overall survival in luminal and HER2 positive tumors. (Supplementary Figure 1).
The text has been modified as follows: “Transcriptomic data from luminal (n=307), HER2+ (n=105) and TNBC (n=97) with a median follow-up of 33 months were used in this analysis. Results showed that low miR-935 levels (HR = 0.25, logrank P=0.02) and high HIF1A expression (HR=1.53, logrank P=0.00076) were associated with poor overall survival of TNBC patients (Figure 9). In contrast, no significant association between miR-935 and HIF1A expression and overall survival in luminal and HER2+ breast tumor subtypes were found (Supplementary Figure 1)”. (Page 12, lanes 460-466).
Minor comments:
- The abstract is too long. It is recommended that the authors to keep the abstract succinct and highlight only the important key messages to the readers.
Reply: As the reviewer suggested, the abstract have been condensed highlighting the key points of the topic.
- Fig 6 is illegible; it is recommended to either reduce the number of hub genes by increasing the stringency of the analysis and to only show key interactions in the main figure.
Reply: Thank you very much for your wise observations. We have reanalyzed and redesigned the figure. Data for the coregulation network was again filtered based on the major signaling pathways (Hypoxia, WNTβ catenin, TGFβ, JAK/STAT3, Cell cycle, Apoptosis, NOTCH, PI3K/AKT/MTOR, P53 and EPITHELIAL/MESENCHYMAL TRANSITION) involved in cancer progression. In the new figure 6 we clearly showed the miRNAs/mRNAs coregulation networks and the cellular processes where they function in colors, making more easy to follow the data for readers (Page 13).
- Throughout the manuscript, the authors used repressed genes to represent downregulated/underexpressed genes. Repressed genes usually refer to switching off of genes. Please clarify and reword the main text if necessary.
Reply: Thank you very much for your very accurate comment, we have revised and corrected the concept throughout the manuscript.
- All genes in main text/ figures should be in italics to differentiate them from proteins.
Reply: Thanks for the observation, we have changed in italics all the genes in the manuscript and figures.
Round 2
Reviewer 2 Report
The authors have addressed most of my concerns except for my cooments on Fig 2. Can the authors show by immunostainings on the mesenchymal markers as described in the text " lamellipodia and filopodia-like structures, corresponding to a mesenchymal phenotype". In addition, the 3D culture of Hs578T in Fig 2 does not look convincing, please include an additional phase contrast photo of the 2D vs 3D micrograph to improve on the description of morphology.
Author Response
We acknowledge to editors and reviewers 2 for the opportunity to revise again the manuscript. Your critical suggestions that we have fully replied greatly increase the quality of our study. All amendments have been marked in yellow color in the revised text for your easy reference and reading. We have carefully reviewed the manuscript according the referee suggestions and provide a point-by-point response. We hope that the manuscript in its actual form fulfill the originality and quality requirements for a Cancers report.
Reviewer 2
The authors have addressed most of my concerns except for my cooments on Fig 2.
Can the authors show by immunostainings on the mesenchymal markers as described in the text " lamellipodia and filopodia-like structures, corresponding to a mesenchymal phenotype".
REPLY: Thanks for the wise comments. The aim of our investigation was the discovery of changes in the microRNAs genomic profiles of cell cultured in 2D and 3D conditions, as well as the elucidation of miRNAs/mRNAs coregulation networks. Our results have implications in our understanding of cancer biology and impacts the miRNAs/mRNAs regulatory axes of cells grown in 3D cultures. Data also represents a guide for novel miRNAs candidates for functional analysis including the response to therapy and biomarkers discovery in breast cancer.
Regarding the comments of reviewer about the immunofluorescence assays showed in figure 2, and to avoid overinterpretation and confusion, we have removed the phrase “corresponding to a mesenchymal phenotype", and we just describe the general morphology of cells in 2D and 3D. Also, we removed the statements about the potential phenotypes and molecular mechanisms potentially associated to the morphological changes observed in the cells, as the characterization of these markers are not the goal of our research.
The experiment suggested by the reviewer about the immunostainings on the mesenchymal markers is largely out of the scope of the present study.
Moreover, too many papers already reported the study of mesenchymal markers for breast cancer 3D cultures (see table 1 and list of papers), thus its characterization in the present study will be, not only redundant, but out of the scope and no original.
Table 1. Biomarkers that have been used to demonstrate EMT in 3D cultures of breast cancer.
|
Biomarkers for EMT |
Expression |
|
E-cadherin |
Downregulated |
|
N-cadherin |
Upregulated |
|
Vimentin |
Upregulated |
|
Snail |
Upregulated |
|
Fibronectin |
Upregulated |
von Nandelstadh, P., et al. (2014). Actin-associated protein palladin promotes tumor cell invasion by linking extracellular matrix degradation to cell cytoskeleton. Molecular biology of the cell, 25(17), 2556–2570. https://doi.org/10.1091/mbc.E13-11-0667
Katz, E., et al. (2011). An in vitro model that recapitulates the epithelial to mesenchymal transition (EMT) in human breast cancer. PloS one, 6(2), e17083. https://doi.org/10.1371/journal.pone.0017083
Kenny, P. A.,et al. (2007). The morphologies of breast cancer cell lines in three-dimensional assays correlate with their profiles of gene expression. Molecular oncology, 1(1), 84–96. https://doi.org/10.1016/j.molonc.2007.02.004
Kai, K., et al. (2018). CSF-1/CSF-1R axis is associated with epithelial/mesenchymal hybrid phenotype in epithelial-like inflammatory breast cancer. Scientific reports, 8(1), 9427. https://doi.org/10.1038/s41598-018-27409-x
Lee, M. S., et al . (2014). Snail1 induced in breast cancer cells in 3D collagen I gel environment suppresses cortactin and impairs effective invadopodia formation. Biochimica et biophysica acta, 1843(9), 2037–2054. https://doi.org/10.1016/j.bbamcr.2014.05.007
Pallegar, N. K., et al. (2019). A Novel 3-Dimensional Co-culture Method Reveals a Partial Mesenchymal to Epithelial Transition in Breast Cancer Cells Induced by Adipocytes. Journal of mammary gland biology and neoplasia, 24(1), 85–97. https://doi.org/10.1007/s10911-018-9420-4
Franchi, M., et al. (2020). Long filopodia and tunneling nanotubes define new phenotypes of breast cancer cells in 3D cultures. Matrix biology plus, 6-7, 100026. https://doi.org/10.1016/j.mbplus.2020.100026
However, we have considered the idea of reviewer about the potential role of mesenchymal markers in the morphological changes of 3D cultures. Inspired in your ideas we have started the characterization of microRNAs modulated in 3D (miR-152-3p, miR-935, miR-25-5p) that potentially regulates several genes involved in the EMT process. This new research project, in progress, will be the main goal in a future paper.
In addition, the 3D culture of Hs578T in Fig 2 does not look convincing, please include an additional phase contrast photo of the 2D vs 3D micrograph to improve on the description of morphology.
REPLY: As the reviewer suggested, we have improved the images to show clearer the existence of lamelipodia and filopodia structures in 2D and 3D cells cultures for easy comprehension of readers. We indicated these structures with differential colored arrows. We add phase contrast images which clearly showed the aforementioned structures.
Round 3
Reviewer 2 Report
The authors have addressed all of my concerns adequately and the quality of the manuscript has greatly improved.